# Predicting the Potential Distribution of *Galeruca daurica* in Inner Mongolia Under Current and Future Climate Scenarios Using the MaxEnt Model

**DOI:** 10.3390/biology14111477

**Published:** 2025-10-23

**Authors:** Tian-Yu Xu, Xiao-Shuan Bai, MU Ren

**Affiliations:** 1College of Life Science and Technology, Inner Mongolia Normal University, Hohhot 010022, China; 19862923622@163.com; 2Key Laboratory of Biodiversity Conservation and Sustainable Utiization, Mongolian iPateal College, University of inner Mongolia Autonomous Region, Hohhot 010022, China

**Keywords:** *Galeruca daurica*, MaxEnt, habitat suitability modeling, climate change, pest risk assessment, Inner Mongolia

## Abstract

**Simple Summary:**

In recent years, the grasslands of Inner Mongolia, an important ecological region in Northern China, have experienced severe outbreaks of a pest beetle known as *Galeruca daurica*. These outbreaks are largely driven by climate change and grassland degradation. To help anticipate future risks, this study used a computer modeling approach to predict where this beetle could live under current and future climate conditions. We found that key factors affecting its distribution are rainfall patterns and temperature variation. At present, about 45% of Inner Mongolia offers a suitable habitat, mainly in the central and western dry grasslands. In the future, however, suitable areas are projected to shrink and shift northward. These results can help guide early pest warnings and focused management efforts, supporting the protection of grassland ecosystems and sustainable land use in the region.

**Abstract:**

In the context of climate change and grassland degradation, the Inner Mongolia Autonomous Region, a key ecological barrier in Northern China, has faced recurrent outbreaks of the pest beetle *Galeruca daurica*. This study aims to project its potential geographic distribution under current and future climate scenarios to support risk assessment and management strategies. Using the Maximum Entropy (MaxEnt) model with 122 occurrence records and environmental variables (climatic, topographic, and edaphic), we simulated habitat suitability under present conditions and future scenarios (SSP1-2.6, SSP2-4.5, and SSP5-8.5 for the 2050s and 2070s). The model performed excellently (AUC > 0.9), with key predictors being precipitation of the wettest month (39.6%), annual precipitation (24.0%), and annual temperature range (8.2%). Currently, about 44.9% of the region is suitable habitat, mainly in central–western arid and semi-arid areas. Future projections indicate a contraction in suitability, which is most pronounced under SSP2-4.5 (declining to 23.56% by the 2070s), along with a northward shift in the distribution centroid. These findings suggest that climate change will likely reduce and shift the suitable range of *G. daurica*, providing a scientific basis for early warning and targeted control in vulnerable grassland ecosystems.

## 1. Introduction

The Inner Mongolia Autonomous Region is characterized by a distinct ecological gradient, with mean annual precipitation decreasing from east to west (150–400 mm) and vegetation transitioning from meadow steppes in the east to desert steppes in the west [1,2,3]. This arid and semi-arid environment provides suitable conditions for grassland pests, while ongoing grassland degradation and desertification further disrupt ecological balance and weaken natural predator control, contributing to frequent pest outbreaks [4,5].

The *Galeruca daurica* belongs to the order Coleoptera, family Chrysomelidae, and subfamily Galerucinae Its larvae primarily feed on the leaves of Allium species (Liliaceae), such as Mongolian onion (*Allium mongolicum*), many-rooted onion (*Allium polyrhizum*), and wild leek (*Allium ramosum*), and may severely damage roots in extreme cases [6]. This pest is native to Mongolia, Russia, North Korea, and South Korea, and is widely distributed in China’s Inner Mongolia, Xinjiang, and Gansu [7]. In recent years, influenced by global warming and grassland degradation, *Galeruca daurica* experienced its first large-scale outbreak in the grasslands of Inner Mongolia in 2019, emerging as a major pastoral pest [8]. Affected regions include Hulunbuir, Xing’an League, Xilingol League, and Ulanqab [8]. Thus, projecting its potential distribution under future climate scenarios is vital for ecological risk early-warning and pest management.

There is a close relationship between species distribution and the environment. Climatic factors have a great influence on species distribution [9]. Elucidating the mechanisms of interaction between organisms and their living environment is an essential prerequisite for understanding the ecological adaptive characteristics of species and the patterns of their geographical distribution [10,11]. The spatial distribution pattern of species is the result of the combined effects of bioclimatic factors, community succession processes, and human disturbance activities. These elements together shape the ecological suitability of habitats and directly affect the species’ potential for continued survival [12]. With the intensification of global warming, many plant and animal species are gradually migrating to higher latitudes or altitudes [13,14]. In response to the impacts of climate change, numerous species are continuously shifting toward habitats with more suitable climatic conditions, leading to dynamic changes in their geographical distribution ranges [15]. Besides climate, topographic factors (e.g., elevational gradients) and soil properties (including physicochemical characteristics) also influence the geographical distribution patterns of insects. Altitudinal variation acts as an ecological barrier by restricting species’ dispersal capacity, while soil pH, organic matter content, and porosity directly affect oviposition site selection, larval development, and microbial symbiosis in insects. Furthermore, human activities such as urbanization and deforestation can significantly alter insect habitats, leading to shifts in their distribution ranges [16]. The combined effects of these factors drive the formation of species’ geographical distribution patterns and their responses to environmental changes.

The Maximum Entropy (MaxEnt) model is widely recognized for its robust predictive performance in species distribution modeling, particularly with limited occurrence data [17,18]. Its principle involves estimating the probability of species occurrence by maximizing entropy subject to environmental constraints derived from known presence locations [19]. When compared to traditional climate-envelope models such as Climax or BIOCLIM, which primarily define species’ climatic tolerances based on bounding boxes and often require both presence and absence data for calibration, the MaxEnt model offers several distinct advantages [20]. Firstly, MaxEnt requires only species presence data and environmental variables, circumventing the challenge of confirming true species absence which is often problematic in field surveys. Secondly, as a machine learning algorithm, it can capture complex non-linear relationships and interactions between environmental predictors, thereby providing a more refined and ecologically realistic estimation of the species’ fundamental niche [21]. In this study, the model was applied to integrate multi-dimensional environmental variables—including climatic, topographic, and notably soil factors—to enhance prediction accuracy. Furthermore, comparative analyses under multiple climate scenarios (SSP1-2.6, SSP2-4.5, and SSP5-8.5) were conducted to project spatiotemporal shifts in habitat suitability [22,23]. This approach offers improved mechanistic insight into the species’ ecological niche and potential range dynamics [24].

This study collected and screened geographical distribution data of *Galeruca daurica*, combined with relevant environmental variables, and employed the Maximum Entropy (MaxEnt) model to predict potential suitable habitats in Inner Mongolia under current and future climate conditions. The research further examined the impact of contemporary and future climate change on habitat distribution, aiming to address the following questions: (1) the potential suitable habitat distribution of *G. daurica* in Inner Mongolia under current climate conditions; (2) changes in potential suitable habitats under future climate scenarios; and (3) the relationship between the beetle’s potential geographical distribution and environmental factors in Inner Mongolia. The findings provide a theoretical foundation and scientific support for early ecological warning and pest control strategies in the region.

Based on the research questions outlined above, this study tests the following central hypotheses: (1) The potential suitable habitat of *Galeruca daurica* under current climatic conditions is primarily restricted to the arid and semi-arid steppe regions of Central Inner Mongolia, constrained by key climatic factors such as precipitation seasonality and temperature range. (2) Under future climate scenarios, the overall extent of suitable habitat will undergo a significant range contraction, accompanied by a pronounced northward shift in the distribution centroid towards higher latitudes.

Ultimately, this research aligns with the United Nations Sustainable Development Goal 15 (Life on Land) by providing a scientific basis for protecting and restoring grassland ecosystems, promoting sustainable land management, and halting biodiversity loss through improved pest monitoring and control strategies [25]. The findings provide a theoretical foundation and scientific support for early ecological warning and pest control strategies in the region.

## 2. Materials and Methods

### 2.1. Species Occurrence Data and Preprocessing

Occurrence records of *Galeruca daurica* were compiled from field surveys conducted during June–August of 2022 and 2023 in Inner Mongolia. The surveys employed line-transect sampling combined with sweep-netting techniques to ensure comprehensive coverage and representative sampling.

Study sites were selected systematically based on two criteria: historical outbreak records of *G. daurica*, and the distribution range of its primary host plants (*Allium mongolicum*, *A. polyrhizum*, and *A. ramosum*). This approach ensured comprehensive coverage across Inner Mongolia’s major grassland types.

To mitigate potential model overfitting caused by spatial autocorrelation, the initial 148 occurrence points were processed using ENM Tools v1.4. A total of 26 spatially redundant points were removed, resulting in 122 unique and spatially independent records retained for model calibration. All geographic coordinates were recorded using handheld GPS devices (Garmin GPSMAP 64s) with ≤5 m accuracy. The device was manufactured by Garmin Ltd., located in Olathe, KS, USA (Appendix A).

### 2.2. Environmental Variables

A total of 36 environmental variables were initially considered, including:19 bioclimatic variables (current and future) from the WorldClim database (https://www.worldclim.org/; accessed on 1 April 2024) under three Shared Socioeconomic Pathways (SSP1-2.6, SSP2-4.5, SSP5-8.5) for the mid-century (2041–2060) and late-century (2061–2080) periods;3 topographic variables (elevation, slope, and aspect);14 soil variables obtained from the Harmonized World Soil Database (HWSD).

All climate data were obtained at a spatial resolution of 2.5 arcminutes.

To reduce multicollinearity among predictors and improve model parsimony, and pairwise Pearson correlations were calculated. If the absolute correlation coefficient between two variables exceeded 0.8, the variable with the lower contribution to the model was excluded. Additionally, only variables with contribution rates ≥0.5% were retained [26,27,28]. Following this two-step screening, 12 environmental predictors were selected for final model construction (Appendix A).

### 2.3. Model Construction and Evaluation

MaxEnt v3.4.4 was employed to model the current and future distributions of *G. daurica*. The 122 occurrence records and 12 selected environmental variables were input into the model [29,30]. Data were randomly partitioned: 75% for training and 25% for testing. The model was set to run for a maximum of 10,000 iterations with 10 replicates. Output was generated in “logistic” format with default settings [31,32].

Model performance was evaluated using the area under the receiver operating characteristic curve (AUC). AUC values > 0.9 indicate excellent predictive performance [33,34]. Variable importance was assessed using the built-in Jackknife test and response curves.

### 2.4. Habitat Suitability Classification

Habitat suitability was classified using the Jenks natural breaks method in ArcGIS. Predicted suitability values (*p*) were categorized as follows:Non-suitable (*p* < 0.1);Low suitability (0.1 ≤ *p* ≤ 0.27);Medium suitability (0.27 < *p* ≤ 0.48);High suitability (*p* > 0.48).

Each habitat class was visualized, and the area of each suitability zone was quantified using the Reclassification and Zonal Statistics tools in ArcGIS 10.8.1 [35,36].

## 3. Results

### 3.1. Model Performance

The MaxEnt model demonstrated robust predictive power under both current and future climate scenarios. All models achieved AUC values exceeding 0.90 (Table 1, Figure 1), indicating high model reliability and accuracy in predicting *G. daurica*’s habitat suitability.

### 3.2. Key Environmental Drivers

Among the 12 environmental predictors, the three most influential were the following:Precipitation of the wettest month (bio13)—39.6% contribution;Annual precipitation (bio12)—24.0%;Annual temperature range (bio7)—8.2%.

These top predictors contributed a cumulative 71.8% to the model. Other relevant variables included subsoil clay content (6.1%) and topsoil sand content (4.8%), which is consistent with the widespread distribution of sandy soils in Central–Western Inner Mongolia. Jackknife analyses further confirmed the critical role of bio13 and bio12, both achieving AUC values > 0.8 when used in isolation (Figure 2, Appendix A).

### 3.3. Response Curve Analysis

Response curves (Figure 3, Appendix A) indicated the optimal ecological ranges for *G. daurica* occurrence:Precipitation of the wettest month: 51–82 mm, with maximum presence probability (0.66) at 62 mm.Annual precipitation: 180–300 mm, optimal at 240 mm.Annual temperature range: 48.5–54.7 °C, optimal at 52 °C.

### 3.4. Current Potential Distribution

Under current climate conditions, *G. daurica*’s suitable habitat in Inner Mongolia encompasses approximately 531,300 km^2^, or 44.9% of the region:High suitability: 117,500 km^2^ (9.9%);Moderate suitability: 149,200 km^2^ (12.6%);Low suitability: 264,600 km^2^ (22.4%).

High-suitability areas are concentrated in the central–western region, notably the Ordos Plateau, Hetao Plain, and parts of Xilingol League and Ulanqab (Figure 4). Field surveys and literature confirm that these regions are characterized by sandy soils and host plant-rich steppe communities dominated by Artemisia ordosica and Stipa bungeana, creating a potentially ideal habitat for *G. daurica* (Appendix A).

### 3.5. Future Habitat Shifts

Climate projections suggest notable spatiotemporal changes in habitat suitability:SSP1-2.6: Slight short-term expansion in the 2050s (835,100 km^2^), followed by a decrease by the 2070s (782,300 km^2^).SSP2-4.5: Steady contraction to 486,300 km^2^ by 2070s (23.56% of Inner Mongolia).SSP5-8.5: An initial decline followed by moderate rebound (801,000 km^2^ in the 2070s).Overall, a northward shift in high-suitability areas is observed under all scenarios, with fragmentation of medium- and low-suitability zones (Figure 5 and Figure 6, Table 2).

### 3.6. Centroid Migration

The current centroid of *G. daurica*’s suitable habitat is located in Sunite Right Banner, Xilingol League (112.7390° E, 43.1234° N). Under future climate scenarios, this centroid is projected to shift northward, with the most pronounced movement (183.6 km) under SSP1-2.6 by the 2050s (Figure 7, Table 3 and Table 4).

## 4. Discussion

Insects, as poikilothermic organisms, have body temperatures that are largely dictated by ambient environmental conditions. Consequently, temperature and moisture availability are key climatic determinants of their development, reproduction, behavior, and survival [37]. Water not only supports basic physiological functions such as metabolism and excretion but also plays a central role in hormone transport and signal transduction [38]. These physiological dependencies render insects highly sensitive to climatic fluctuations [39]. The combined effects of temperature and precipitation operate via both direct (physiological) and indirect (habitat structural) mechanisms, ultimately shaping habitat suitability and determining the spatial distribution of insect populations [40,41,42].

Our results confirm that temperature and precipitation are the most critical environmental variables governing the distribution of *Galeruca daurica*. Among the 12 environmental predictors used in the MaxEnt model, precipitation of the wettest month (bio13), annual precipitation (bio12), and annual temperature range (bio7) together accounted for 71.8% of the cumulative contribution to habitat suitability. These findings are consistent with previous studies and highlight *G. daurica*’s strong ecological affinity for semi-arid to arid climates [43]. Response curve analysis further revealed that the beetle favors environments with lower precipitation levels. This preference aligns with the known susceptibility of many soil-dwelling insect larvae and pupae to conditions of high humidity, where excessive moisture can disrupt cuticular respiration, promote pathogenic fungal growth, and increase mortality rates [44]. Consequently, the arid and semi-arid conditions predicted to be suitable for *G. daurica* are likely to reduce such physiological stresses during its critical underground developmental stages. Additionally, the suitable annual temperature range for the beetle’s development was estimated at 48.5–54.7 °C (bio7), a range known to influence key life-history traits such as larval development rate and diapause termination.

Beyond the dominant climatic controls, our model also identified soil texture as a significant predictor, highlighting the importance of local habitat conditions. Our model identified soil texture (clay and sand content) as a significant predictor, which can be explained by its direct and indirect effects on *G. daurica*. The sandy soils prevalent in the high-suitability areas of the Ordos Plateau and Hetao Plain [45] offer well-drained conditions that are crucial for the survival of *G. daurica*’s soil-dwelling life stages (e.g., eggs and pupae) [6]. Moreover, these soil conditions structure the plant community, favoring the growth of its host plants, such as Allium mongolicum, within a xerophytic vegetation matrix dominated by Artemisia ordosica and Stipa bungeana [46]. Therefore, the distribution of *G. daurica* appears to be a function of not only macro-climate but also locally filtered conditions where suitable soils support the necessary host plant communities, creating a hierarchy of habitat suitability.

Climatic variables also exert indirect control over beetle distribution by influencing the phenology, nutritional quality, and spatial availability of host plants, primarily *Allium mongolicum* and related species. Previous studies have shown that temperature and precipitation changes affect host plant biomass and chemistry, which in turn modulate herbivore fitness, survival, and dispersal capacity [47,48,49]. Therefore, shifts in the climatic niche of *G. daurica* may reflect both direct thermohydric limitations [50] and indirect host-plant-mediated constraints [51].

Under current climate conditions, *G. daurica*’s high-suitability habitats are concentrated in the temperate arid and semi-arid belt between 110° E and 120° E. These regions are characterized by abundant solar radiation, sandy soils, sparse vegetation structure, and low annual precipitation—conditions that support the species’ xerophytic life-history strategy. Central Inner Mongolia, including parts of the Ordos Plateau and Xilingol League, hosts contiguous high-suitability zones due to the co-occurrence of favorable abiotic factors and abundant host plant resources. In contrast, regions west of 100°E are constrained by higher soil clay content and slightly increased precipitation, while high-latitude zones (near 50° N) remain largely unsuitable due to thermal limitations.

Climate change projections reveal a general contraction of suitable habitat area for *Galeruca daurica*, particularly under moderate- to high-emission scenarios (SSP2-4.5 and SSP5-8.5). This contraction is driven by both climatic mechanisms and exacerbated by current grassland utilization patterns. Specifically, the following processes interact with regional grazing intensity and degradation-recovery dynamics:

(1) Increased precipitation: Projected rises in precipitation exceed the species’ optimal hydrological thresholds, especially during the wettest month, reducing habitat suitability. This effect may be intensified in overgrazed or degraded grasslands, where reduced vegetation cover and compromised soil structure alter water infiltration and retention, further disrupting the microhabitat conditions required by *G. daurica*.

(2) Elevated temperatures: Warmer conditions disrupt critical physiological processes such as diapause termination, which depends on sufficient cold accumulation. In degraded grasslands with reduced plant biomass and lower litter cover, thermal buffering capacity is diminished, potentially amplifying temperature stress and impairing embryonic development. Consequently, range persistence may be further limited in areas subject to high grazing pressure and slow ecological recovery.

These findings highlight the importance of integrating grassland management practices—such as restoring degraded areas and regulating grazing intensity—into climate adaptation strategies to mitigate future habitat loss and suppress potential pest outbreaks.

Geographically, our analysis suggests a shift characterized by northward migration and inland westward expansion, with contraction in the southern and eastern regions. Notably, western regions such as the Alxa Plateau and western Bayannur—traditionally arid—may become newly suitable habitats due to intensified aridification under future climate regimes. This finding is consistent with ecological niche theory, which predicts poleward or altitudinal range shifts under warming scenarios, as evidenced by studies on diverse taxa including trees and krill [52]. Critically, this projected westward and northward shift indicates a potential increase in pest invasion risk for adjacent regions in China, including Gansu, Ningxia, and Northern Xinjiang, as well as for the bordering territories of Mongolia. This underscores the necessity for enhanced cross-regional and international monitoring and cooperative management strategies. Conversely, the eastern belt, despite historically supporting large populations, may experience significant suitability loss due to altered precipitation regimes and intensified land use activities [53].

Under all SSP scenarios, the centroid of suitable habitats shifts toward higher latitudes, supporting the hypothesis of climate-driven redistribution. Particularly under SSP2-4.5 and SSP5-8.5, habitat fragmentation and reduced connectivity may pose additional risks to population persistence. These findings underscore the necessity for region-specific pest risk assessments and adaptive management plans to effectively address the spatially heterogeneous impacts of climate change on *Galeruca daurica* distribution [54].

It is recommended that ecological monitoring and early-warning systems be enhanced in newly emerging high-risk areas, particularly in western Inner Mongolia. Simultaneously, ongoing habitat degradation and potential pest collapse in eastern regions require sustained attention to avoid ecosystem-level instability due to secondary pest outbreaks or trophic imbalances.

## 5. Conclusions

This study employed the MaxEnt model to predict the current and future distribution of *Galeruca daurica* in Inner Mongolia. The results demonstrate that suitable habitats for this species is mainly influenced by precipitation during the wettest month, annual precipitation, and annual temperature range. Currently, approximately 44.9% of the region constitutes to suitable habitat, concentrated primarily in the central–western arid and semi-arid grasslands.

Future projections indicate a general contraction in suitable areas, with the most significant decline under the SSP2-4.5 scenario (to 23.56% by the 2070s), along with a consistent northward shift in the distribution centroid.

The modeling approach and environmental variables used here are also applicable to other arid and semi-arid grassland regions with similar ecological conditions, such as parts of Gansu, Ningxia, Xinjiang, and Mongolia, supporting cross-regional pest risk assessment under climate change.

These findings provide a scientific basis for monitoring, early warning, and targeted control of *G. daurica*, and suggest the need to incorporate biotic and anthropogenic factors in future research for improved prediction.

## Figures and Tables

**Figure 1 biology-14-01477-f001:**
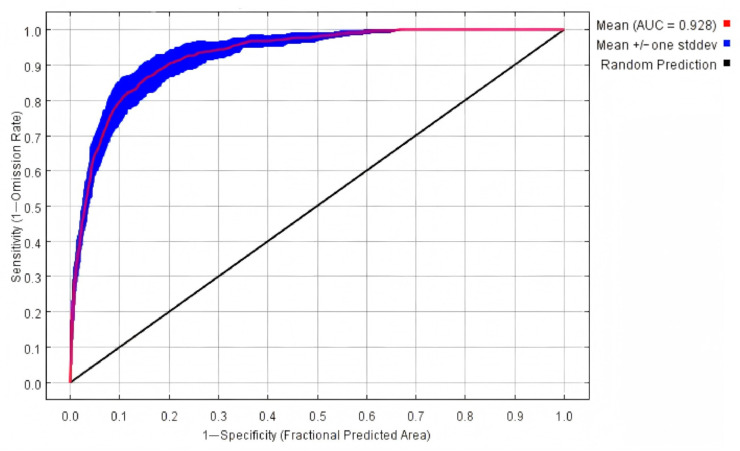
ROC curve of MaxEnt-predicted distribution for *Galeruca daurica*.

**Figure 2 biology-14-01477-f002:**
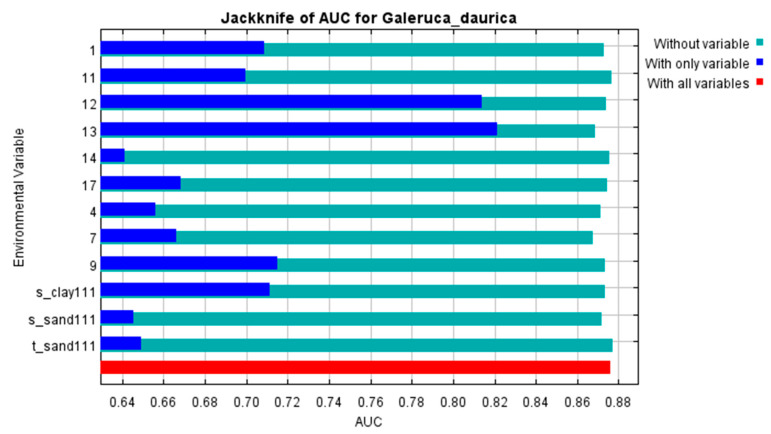
Jackknife test of the contributions of key environmental factors.

**Figure 3 biology-14-01477-f003:**
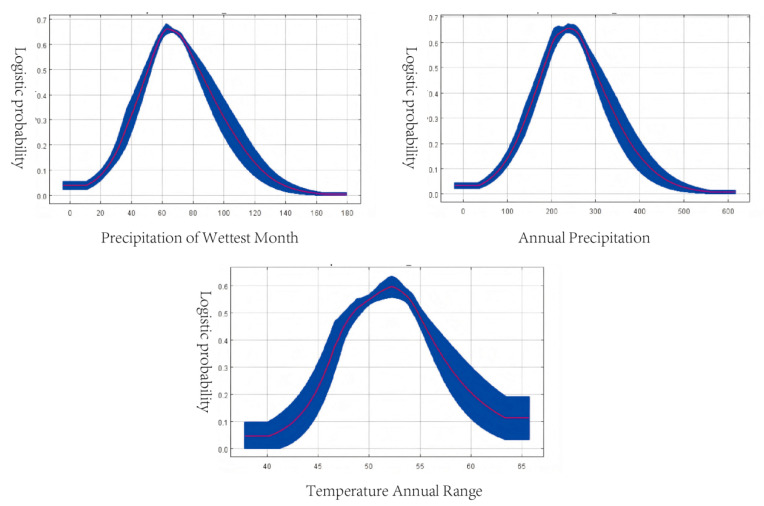
Response curves of key environmental variables.

**Figure 4 biology-14-01477-f004:**
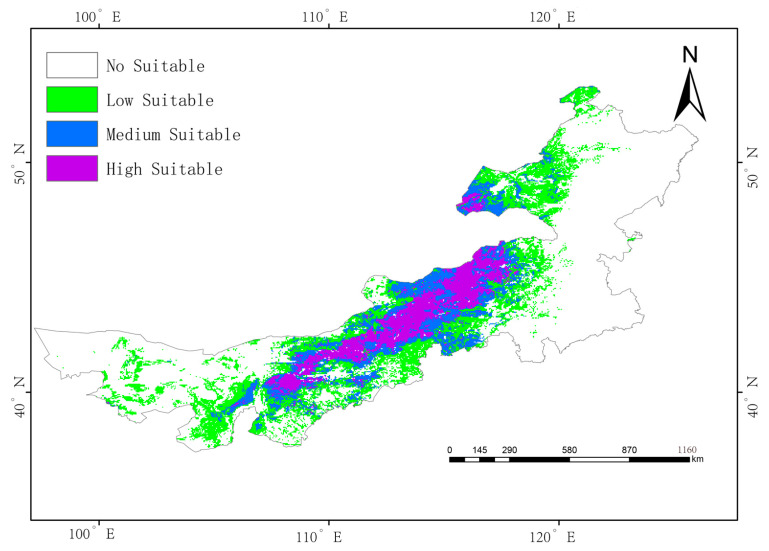
Potential suitable habitat distribution of *Galeruca daurica* in Inner Mongolia under current climatic conditions.

**Figure 5 biology-14-01477-f005:**
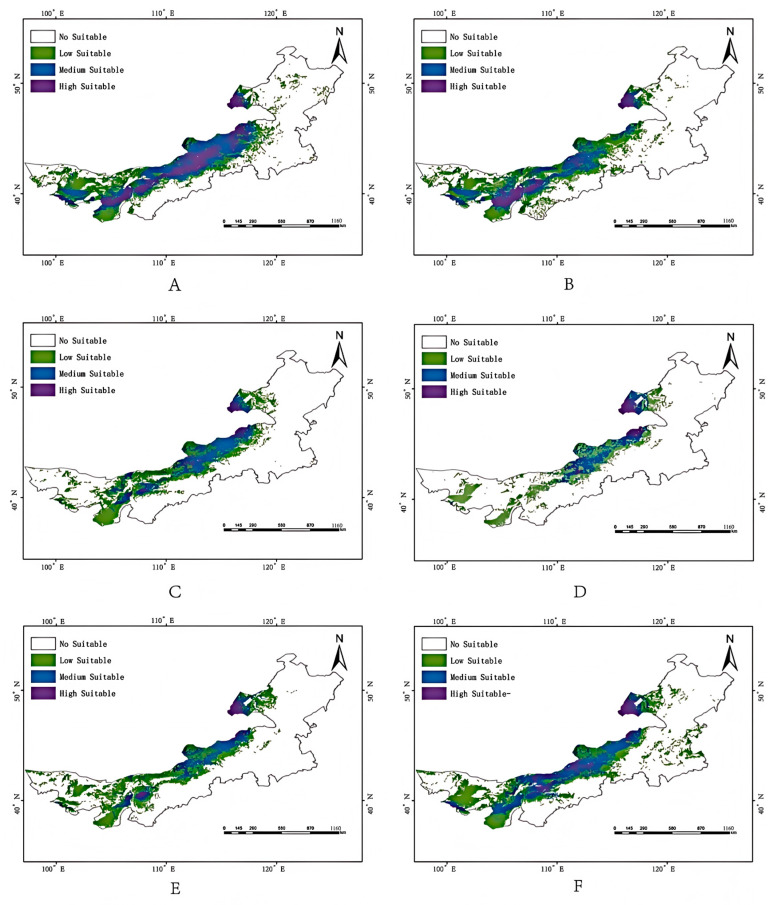
Potential suitable habitat distribution of *Galeruca daurica* in Inner Mongolia under future climate scenarios. (**A**,**B**) SSP1-2.6 for the 2050s and 2070s, respectively; (**C**,**D**) SSP2-4.5 for the 2050s and 2070s, respectively; (**E**,**F**) SSP5-8.5 for the 2050s and 2070s, respectively.

**Figure 6 biology-14-01477-f006:**
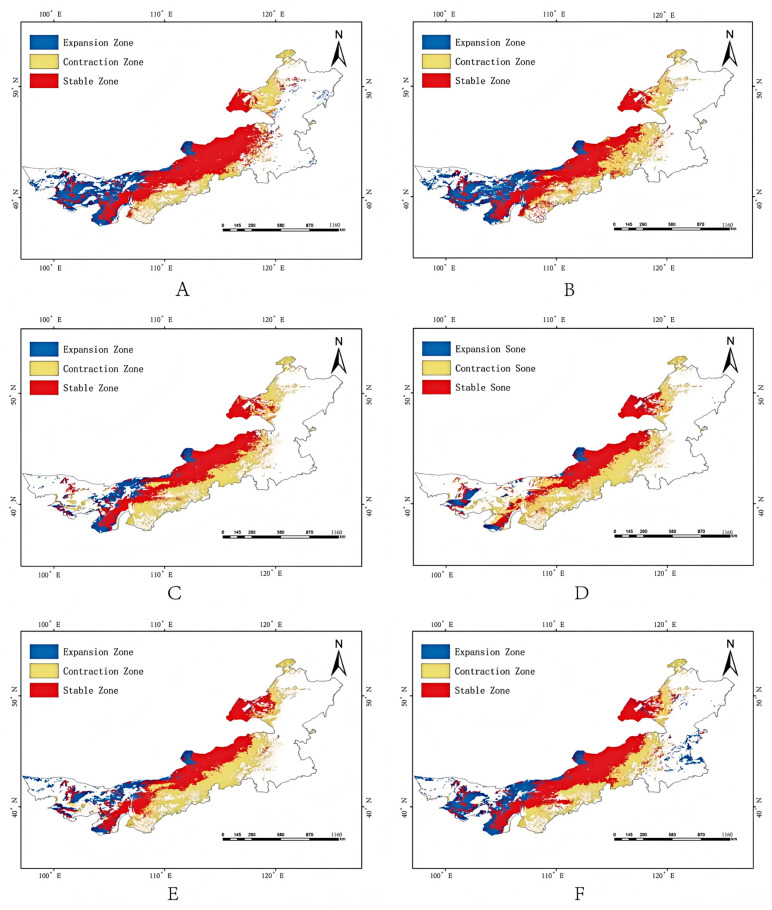
Spatial changes in the suitable habitat of *Galeruca daurica* under future climate scenarios compared to the current distribution. (**A**,**B**) SSP1-2.6 for the 2050s and 2070s, respectively; (**C**,**D**) SSP2-4.5 for the 2050s and 2070s, respectively; (**E**,**F**) SSP5-8.5 for the 2050s and 2070s, respectively.

**Figure 7 biology-14-01477-f007:**
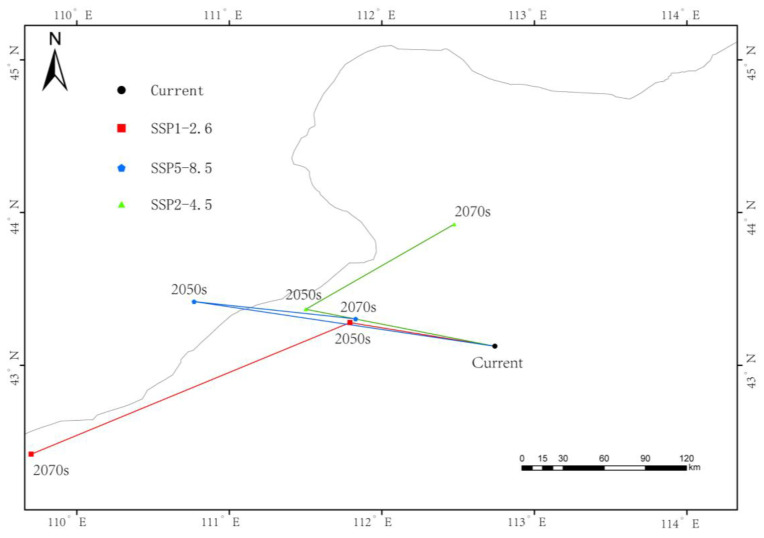
Centroid shift trajectory of *Galeruca daurica* suitable habitats under future climate scenarios.

**Table 1 biology-14-01477-t001:** Description of environmental variables.

Code	Environmental Variables	Unit
Bio1	Annual Mean Temperature	°C
Bio4	Temperature Seasonality	°C
Bio7	Annual Temperature Range	°C
Bio9	Mean Temperature of Driest Quarter	°C
Bio11	Mean Temperature of Coldest Quarter	°C
Bio12	Annual Precipitation	mm
Bio13	Precipitation of Wettest Month	mm
Bio14	Precipitation of Driest Month	mm
Bio17	Precipitation of Driest Quarter	mm
s_sand	Subsoil Sand Content	% weight
s_clay	Subsoil Clay Content	% weight
t_sand	Topsoil Sand Content	% weight

**Table 2 biology-14-01477-t002:** AUC values of MaxEnt-predicted distribution for *Galeruca daurica*.

	Current	2050s			2070s		
Climate Scenario	—	SSP1-2.6	SSP2-4.5	SSP5-8.5	SSP1-2.6	SSP2-4.5	SSP5-8.5
AUC	0.928	0.924	0.928	0.930	0.933	0.923	0.921

**Table 3 biology-14-01477-t003:** Suitable habitat area of *Galeruca daurica* under current and future climate conditions.

Circumstances	Low Suitable	Medium Suitable	High Suitable	All
Current	26.45	14.92	11.76	53.13
2050-126	21.05	16.41	14.35	51.81
2050-245	20.61	11.65	3.59	35.85
2050-580	23.12	9.17	3.3	35.59
2070-126	25.36	13.26	6.91	45.53
2070-245	17.67	6.99	3.17	27.83
2070-585	24.31	15.35	6.84	46.5

Unit: ×10^4^ km^2.^

**Table 4 biology-14-01477-t004:** Centroid coordinates and shift distances of *Galeruca daurica* suitable habitats under future climate scenarios.

Stage	Longitude	Latitude	Migration Distance (km)
SSP1-2.6-2050s	111.7907	43.2791	183.64
SSP1-2.6-2070s	109.7002	42.4163	27.27
SSP2-4.5-2050s	111.5007	43.3661	112.73
SSP2-4.5-2070s	112.4735	43.9228	47.27
SSP5-8.5-2050s	110.7691	43.4158	122.73
SSP5-8.5-2070s	111.8278	43.3025	121.82

## Data Availability

All raw data from the research process will be provided upon request.

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
