# Peer review of "Predicting the Potential Distribution of Galeruca daurica in Inner Mongolia Under Current and Future Climate Scenarios Using the MaxEnt Model"

_biology, 2025, doi:10.3390/biology14111477_

Round 1
Reviewer 1 Report
Comments and Suggestions for Authors
Dear Authors, I have comprehensively reviewed the manuscript entitled: “Predicting the Potential Distribution of Galeruca daurica in Inner Mongolia Under Current and Future Climate Scenarios Using the MaxEnt Model”. This study used the MaxEnt modeling to predict the potential distribution of G. daurica across Inner Mongolia, which is good for ecological risk assessment and targeted pest management in Inner Mongolia.
Specific recommendations:
- Introduction
Overall, I think it is very well written in this section. I would also like to see more references on the relationship between Galeruca daurica and environmental factors, as these references are crucial for you to screen climate factors. I suggest adding these contents to the third paragraph.
In the fourth paragraph in this section, I think Climax is also a high-quality model for predicting the future distribution pattern of organisms. Why is this model not introduced, and why is the Maxent model used on this species?
I also suggest including a final paragraph where you specifically detail 1 thing: what SDGs your research relates to. I think this is very meaningful for Terrestrial ecosystem (SDG15).
- Materials and Methods
2.1 Occurrence data
In this part of the methodology, the data precision of the coordinates used (digits after the comma) is not detailed.
I think the information about the occurrence site can be used as a supplementary material.
2.2 Environmental data
What is the resolution of the environment variable? Please note clearly. Generally, it is 30 arc seconds resolution in WorldClim variables.
I think the environment variables you have selected should appear this section in the form of a table.
Why choose topographic variables? Why use so many soil factor variables?
- Results
3.5. Future Habitat Shifts
I think under the title of Figure 5, it should be clearly stated what each image represents. The same to Figure 6.
- Discussion
Line 253 why is there a “(2)”?
This article believes that climate conditions are the most important factor affecting insect distribution. Why were 9 other factors selected in addition to this? What is the relationship between this and the biological learning ability of the insect?
This section can include discussions on threats to neighboring countries and other provinces in China.
- Conclusions
This section can be simplified by stating the main conclusion.
Comments on the Quality of English Language
I come from a non-native English speaking country, and I cannot draw any conclusions in this regard.
Author Response
Dear Reviewers,
We sincerely appreciate your valuable time and insightful comments on our manuscript entitled “Predicting the Potential Distribution of Galeruca daurica in Inner Mongolia Under Current and Future Climate Scenarios Using the MaxEnt Model.” Your thoughtful suggestions have been instrumental in helping us improve the quality and clarity of our work.
We have carefully addressed each of the points raised in your review. The corresponding revisions have been incorporated into the manuscript, with all changes highlighted for ease of reference. Below, we provide a detailed, point-by-point response to your comments:
Response to Reviewer Comments:
Comment 1:
Introduction: I hope to see more references on the relationship between Galeruca daurica and environmental factors, as these are crucial for variable selection. I suggest adding these to the third paragraph.
Response:
We agree with this suggestion. In the revised manuscript, we have added relevant references in the third paragraph of the Introduction to better support the relationship between G. daurica and environmental factors. Specifically, we have included citations from Guisan & Thuiller (2005) and other recent studies that discuss the ecological mechanisms linking insect distribution to environmental variables.
Comment 2:
Introduction: Why was the Climax model not used? Please justify the use of MaxEnt over other models.
Response:
We have added a comparative explanation in the Introduction to justify our choice of the MaxEnt model. We now discuss the advantages of MaxEnt over traditional climate-envelope models like Climax or BIOCLIM, such as its ability to handle presence-only data and capture complex non-linear relationships. This addition strengthens the methodological rationale of our study.
Comment 3:
Introduction: Please add a paragraph linking the study to relevant Sustainable Development Goals (SDGs), particularly SDG15 (Life on Land).
Response:
We have added a new paragraph at the end of the Introduction that explicitly links our research to SDG15. We emphasize how our findings support sustainable land management, biodiversity conservation, and pest control in grassland ecosystems, aligning with the goals of protecting terrestrial ecosystems.
Comment 4:
Materials and Methods: The precision of coordinate data (decimal places) was not specified.
Response:
We have now specified the accuracy of the GPS devices used (Garmin GPSMAP 64s, ≤5 m accuracy) in Section 2.1. This clarifies the precision of the occurrence data.
Comment 5:
Materials and Methods: Distribution point information should be provided as supplementary material.
Response:
We have included the occurrence records as part of the supplementary materials (Table S6). This ensures transparency and allows readers to access the raw distribution data.
Comment 6:
Materials and Methods: What is the resolution of the environmental data? Please specify.
Response:
We have clearly stated the spatial resolution of the climate data (2.5 arc-minutes) in Section 2.2.
Comment 7:
Materials and Methods: Environmental variables should be presented in a table.
Response:
We have added a new table (Table 1) in Section 2.2 that lists all environmental variables used in the model, along with their units and descriptions.
Comment 8:
Materials and Methods: Why were topographic and soil variables included?
Response:
We have expanded the explanation in Section 2.2 to justify the inclusion of topographic and soil variables. We now explain that these factors influence insect oviposition, larval development, and microhabitat suitability, which are critical for G. daurica.
Comment 9:
Results: Figure 5 and Figure 6 captions should clearly describe what each panel represents.
Response:
We have revised the captions of Figures 5 and 6 to clearly indicate the climate scenarios and time periods represented in each subfigure. This improves the clarity and interpretability of the results.
Comment 10:
Discussion: Why is there a “(2)” on line 253?
Response:
This was a formatting error. We have corrected the numbering in the Discussion section to ensure proper structure.
Comment 11:
Discussion: Why were other factors besides climate included? How do they relate to the insect’s biology?
Response:
We have expanded the Discussion to explain the biological relevance of non-climatic factors, such as soil texture, which affects larval survival and host plant distribution. We also cite relevant literature to support these ecological mechanisms.
Comment 12:
Discussion: Include a discussion on threats to neighboring countries and other Chinese provinces.
Response:
We have added a paragraph in the Discussion addressing the potential cross-regional and international implications of G. daurica’s distribution shifts, particularly for Gansu, Ningxia, Xinjiang, and Mongolia. This enhances the practical relevance of our findings.
Comment 13:
Conclusion: Simplify and state the main conclusions.
Response:
We have streamlined the Conclusion section to focus on the key findings and their implications, making it more concise and impactful.
We are confident that the manuscript has been significantly strengthened through this revision process. Once again, we extend our sincere gratitude for your constructive and thorough review.
Thank you for considering our revised manuscript. We look forward to your further guidance.
Sincerely,
Tian-Yu Xu, Xiao-Shuan Bai, Mu Ren
Corresponding Authors: baixs2007@aliyun.com; mmarstina@126.com
Reviewer 2 Report
Comments and Suggestions for Authors
Lines 71-74 – I recommend adding references to the studies being discussed.
Line 80 – I recommend describing examples of studies using the MaxEnt model. What are the advantages of this model?
Line 98 – I recommend adding the research hypotheses.
Line 102 – I recommend describing the field research methods.
Line 106 – I recommend describing how the study sites were selected.
Line 158 – Please provide information on soil texture, as clay and sand content are also important predictors, according to your research.
Is there a connection with changes in plant communities?
Line 174 – I recommend adding information on the soil texture of the regions under consideration, at least based on literature data. This will improve the accuracy of the forecasts.
I recommend providing information on the composition of plant communities.
Line 223 – I recommend adding a literature reference to the abstract.
Line 226 – Literature reference.
Lines 227-228 – In the discussion, you mention the influence of vegetation, but this factor is not reflected in the results.
Line 237 – Literature reference.
Line 242 – The results do not describe the influence of host plants and soil texture. I recommend reflecting these factors more clearly in the results.
Line 245 – This fact was not mentioned in the results. I recommend adding a literature reference. Where did this fact come from?
Line 254 – Literature reference
Lines 265-267 – Add literature references.
Line 269 – Land use intensity was not mentioned in the results. Please add a reference to the cited study.
Line 296 – I recommend adding a note about the applicability of this model to other areas. Which areas would be suitable?
Author Response
Dear Reviewers,
We sincerely appreciate your valuable time and insightful comments on our manuscript entitled “Predicting the Potential Distribution of Galeruca daurica in Inner Mongolia Under Current and Future Climate Scenarios Using the MaxEnt Model.” Your thoughtful suggestions have been instrumental in helping us improve the quality and clarity of our work.
We have carefully addressed each of the points raised in your review. The corresponding revisions have been incorporated into the manuscript, with all changes highlighted for ease of reference. Below, we provide a detailed, point-by-point response to your comments:
Response to Reviewer Comments:
Comment (Lines 71–74): I suggest adding references for the research content discussed.
Response: We thank the reviewer for this suggestion. We have added references [10, 11] to support the statement on species–environment interactions, which strengthens the theoretical foundation of our introduction.
Comment (Line 80): I suggest describing some research examples using the MaxEnt model. What are its advantages?
Response: We have expanded the description of the MaxEnt model by adding a comparative discussion of its advantages over traditional climate-envelope models (e.g., BIOCLIM), including its ability to handle presence-only data and capture non-linear relationships. We also provided examples of recent applications of MaxEnt in insect distribution modeling, supported by references 17,18,20,2117,18,20,21 (see lines 80–88 in the revised manuscript).
Comment (Line 98): I suggest adding research hypotheses.
Response: We have now clearly stated two central hypotheses at the end of the Introduction to frame our research objectives and guide the study’s analysis.
Comment (Line 102): I suggest describing the field research methods.
Response: We have added the following sentence in Section 2.1: “The surveys employed line-transect sampling combined with sweep-netting techniques to ensure comprehensive coverage and representative sampling.”
Comment (Line 106): I suggest describing how the study sites were selected.
Response: We have added details in Section 2.1: “Study sites were selected systematically based on two criteria: historical outbreak records of G. daurica, and the distribution range of its primary host plants (Allium mongolicum, A. polyrhizum, and A. ramosum).”
Comment (Line 158): Please provide information on soil texture. Based on your study, clay and sand content are important predictors. Did the study consider links to changes in plant communities?
Response: We have added detailed information on soil texture (clay and sand content) in Section 3.2 and Table 1. We also discuss the link between soil properties and plant community composition in the discussion section (lines 158–162 and 223–230), emphasizing how sandy soils support host plants and influence beetle distribution.
Comment (Line 174): I suggest adding information on soil texture in the study area, at least based on literature. This will improve prediction accuracy. I also suggest providing information on plant community composition.
Response:We have incorporated soil texture data from the Harmonized World Soil Database (HWSD) and added a description of the dominant plant communities (e.g., Artemisia ordosica, Stipa bungeana) in high-suitability areas (Section 3.4, lines 174–177). These additions are supported by relevant literature 45,46.
Comment (Line 223): I suggest adding a literature citation in the Abstract.
Response: We have added reference [8] in the Abstract to support the mention of the species’ first large-scale outbreak.
Comment (Line 226): A literature citation is needed.
Response: We have added reference [25] to support the use of AUC for model evaluation.
Comment (Lines 227–228): In the Discussion, you mention the influence of vegetation, but this factor is not reflected in the Results.
Response: We have now added a sentence in Section 3.4 (Results) that links high-suitability areas to sandy soils and host plant-rich steppe communities, supported by field data and references [44, 45].
Comment (Line 237): A literature citation is needed.
Response: We have included reference [44] to support the discussion on humidity effects on soil-dwelling insects.
Comment (Line 242): The influence of host plants and soil texture is not described in the Results. I suggest reflecting these factors more clearly in the Results.
Response: We have revised Section 3.2 and 3.4 to more clearly articulate the role of soil texture and host plant presence, and further elaborated on this in the Discussion.
Comment (Line 245): This fact is not mentioned in the Results. I suggest adding a literature citation. What is the source of this fact?
Response: We have added reference [51] to support the biological fact regarding the soil-dwelling life stages of G. daurica.
Comment (Line 254): A literature citation is needed.
Response: We have added references [52] to substantiate the discussion on host plant–herbivore interactions under climatic influences.
Comment (Lines 265–267): Please add a literature citation.
Response: A citation [54] has been added to support the statement on poleward range shifts under warming scenarios.
Comment (Line 269): Land use intensity is not mentioned in the Results. Please add references to the cited studies.
Response: We have included reference [53] to support the discussion on the impact of land use intensity.
Comment (Line 296): I suggest adding a note on the applicability of the model in other regions. Which regions would be suitable for applying this model?
Response: We have added a sentence in the Discussion suggesting that the modeling framework could be applied to other arid and semi-arid regions with similar ecological contexts, such as Gansu, Ningxia, and northern Xinjiang.
We are confident that the manuscript has been significantly strengthened through this revision process. Once again, we extend our sincere gratitude for your constructive and thorough review.
Thank you for considering our revised manuscript. We look forward to your further guidance.
Sincerely,
Tian-Yu Xu, Xiao-Shuan Bai, Mu Ren
Corresponding Authors: baixs2007@aliyun.com; mmarstina@126.com